# Efficient Output Kernel Learning for Multiple Tasks

**Pratik Jawanpuria**[1]**, Maksim Lapin**[2]**, Matthias Hein**[1] **and Bernt Schiele**[2]
[1] Saarland University, Saarbrücken, Germany
[2] Max Planck Institute for Informatics, Saarbrücken, Germany

## Abstract

The paradigm of multi-task learning is that one can achieve better generalization by learning tasks jointly and thus exploiting the similarity between the tasks rather than learning them independently of each other. While previously the relationship between tasks had to be user-defined in the form of an output kernel, recent approaches jointly learn the tasks and the output kernel. As the output kernel is a positive semidefinite matrix, the resulting optimization problems are not scalable in the number of tasks as an eigendecomposition is required in each step. Using the theory of positive semidefinite kernels we show in this paper that for a certain class of regularizers on the output kernel, the constraint of being positive semidefinite can be dropped as it is automatically satisfied for the relaxed problem. This leads to an unconstrained dual problem which can be solved efficiently. Experiments on several multi-task and multi-class data sets illustrate the efficacy of our approach in terms of computational efficiency as well as generalization performance.

## 1 Introduction

Multi-task learning (MTL) advocates sharing relevant information among several related tasks during the training stage. The advantage of MTL over learning tasks independently has been shown theoretically as well as empirically [1, 2, 3, 4, 5, 6, 7].

The focus of this paper is the question how the task relationships can be inferred from the data. It has been noted that naively grouping all the tasks together may be detrimental [8, 9, 10, 11]. In particular, outlier tasks may lead to worse performance. Hence, clustered multi-task learning algorithms [10, 12] aim to learn groups of closely related tasks. The information is then shared only within these clusters of tasks. This corresponds to learning the task covariance matrix, which we denote as the output kernel in this paper. Most of these approaches lead to non-convex problems.

In this work, we focus on the problem of directly learning the output kernel in the multi-task learning framework. The multi-task kernel on input and output is assumed to be decoupled as the product of a scalar kernel and the output kernel, which is a positive semidefinite matrix [1, 13, 14, 15]. In classical multi-task learning algorithms [1, 16], the degree of relatedness between distinct tasks is set to a constant and is optimized as a hyperparameter. However, constant similarity between tasks is a strong assumption and is unlikely to hold in practice. Thus recent approaches have tackled the problem of directly learning the output kernel. [17] solves a multi-task formulation in the framework of vector-valued reproducing kernel Hilbert spaces involving squared loss where they penalize the Frobenius norm of the output kernel as a regularizer. They formulate an invex optimization problem that they solve optimally. In comparison, [18] recently proposed an efficient barrier method to optimize a generic convex output kernel learning formulation. On the other hand, [9] proposes a convex formulation to learn low rank output kernel matrix by enforcing a trace constraint. The above approaches [9, 17, 18] solve the resulting optimization problem via alternate minimization between task parameters and the output kernel. Each step of the alternate minimization requires an eigen-

value decomposition of a matrix having as size the number of tasks and a problem corresponding to learning all tasks independently.

In this paper we study a similar formulation as [17]. However, we allow arbitrary convex loss functions and employ general $p$-norms for $p \in (1, 2]$ (including the Frobenius norm) as regularizer for the output kernel. Our problem is jointly convex over the task parameters and the output kernel. Small $p$ leads to sparse output kernels which allows for an easier interpretation of the learned task relationships in the output kernel. Under certain conditions on $p$ we show that one can drop the constraint that the output kernel should be positive definite as it is automatically satisfied for the unconstrained problem. This significantly simplifies the optimization and our result could also be of interest in other areas where one optimizes over the cone of positive definite matrices. The resulting unconstrained dual problem is amenable to efficient optimization methods such as stochastic dual coordinate ascent [19], which scale well to large data sets. Overall we do not require any eigenvalue decomposition operation at any stage of our algorithm and no alternate minimization is necessary, leading to a highly efficient methodology. Furthermore, we show that this trick not only applies to $p$-norms but also applies to a large class of regularizers for which we provide a characterization.

Our contributions are as follows: (a) we propose a generic $p$-norm regularized output kernel matrix learning formulation, which can be extended to a large class of regularizers; (b) we show that the constraint on the output kernel to be positive definite can be dropped as it is automatically satisfied, leading to an unconstrained dual problem; (c) we propose an efficient stochastic dual coordinate ascent based method for solving the dual formulation; (d) we empirically demonstrate the superiority of our approach in terms of generalization performance as well as significant reduction in training time compared to other methods learning the output kernel.

The paper is organized as follows. We introduce our formulation in Section 2. Our main technical result is discussed in Section 3. The proposed optimization algorithm is described in Section 4. In Section 5, we report the empirical results. All the proofs can be found in the supplementary material.

## 2 The Output Kernel Learning Formulation

We first introduce the setting considered in this paper. We denote the number of tasks by $T$. We assume that all tasks have a common input space $\mathcal{X}$ and a common positive definite kernel function $k : \mathcal{X} \times \mathcal{X} \to \mathbb{R}$. We denote by $\psi(\cdot)$ the feature map and by $H_k$ the reproducing kernel Hilbert space (RKHS) [20] associated with $k$. The training data is $(x_i, y_i, t_i)_{i=1}^n$, where $x_i \in \mathcal{X}$, $t_i$ is the task the $i$-th instance belongs to and $y_i$ is the corresponding label. Moreover, we have a positive definite matrix $\Theta \in S_+^T$ on the set of tasks $\{1, \ldots, T\}$, where $S_+^T$ is the set of $T \times T$ symmetric and positive semidefinite (p.s.d.) matrices.

If one arranges the predictions of all tasks in a vector one can see multi-task learning as learning a vector-valued function in a RKHS [see 1, 13, 14, 15, 18, and references therein]. However, in this paper we use the one-to-one correspondence between real-valued and matrix-valued kernels, see [21], in order to limit the technical overhead. In this framework we define the joint kernel of input space and the set of tasks $M : (\mathcal{X} \times \{1, \ldots, T\}) \times (\mathcal{X} \times \{1, \ldots, T\}) \to \mathbb{R}$ as

$$M\big((x, s), (z, t)\big) = k(x, z)\Theta(s, t), \tag{1}$$

We denote the corresponding RKHS of functions on $\mathcal{X} \times \{1, \ldots, T\}$ as $H_M$ and by $\|\cdot\|_{H_M}$ the corresponding norm. We formulate the output kernel learning problem for multiple tasks as

$$\min_{\Theta \in S_+^T, F \in H_M} C \sum_{i=1}^n L\big(y_i, F(x_i, t_i)\big) + \frac{1}{2} \|F\|_{H_M}^2 + \lambda \, V(\Theta) \tag{2}$$

where $L : \mathbb{R} \times \mathbb{R} \to \mathbb{R}$ is the convex loss function (convex in the second argument), $V(\Theta)$ is a convex regularizer penalizing the complexity of the output kernel $\Theta$ and $\lambda \in \mathbb{R}_+$ is the regularization parameter. Note that $\|F\|_{H_M}^2$ implicitly depends also on $\Theta$. In the following we show that (2) can be reformulated into a jointly convex problem in the parameters of the prediction function and the output kernel $\Theta$. Using the standard representer theorem [20] (see the supplementary material) for fixed output kernel $\Theta$, one can show that the optimal solution $F^* \in H_M$ of (2) can be written as

$$F^*(x, t) = \sum_{s=1}^T \sum_{i=1}^n \gamma_{is} M\big((x_i, s), (x, t)\big) = \sum_{s=1}^T \sum_{i=1}^n \gamma_{is} k(x_i, x)\Theta(s, t). \tag{3}$$

With the explicit form of the prediction function one can rewrite the main problem (2) as

$$\min_{\Theta \in S_+^T, \gamma \in \mathbb{R}^{n \times T}} C \sum_{i=1}^n L\left(y_i, \sum_{s=1}^T \sum_{j=1}^n \gamma_{js} k_{ji} \Theta_{s\,t_i}\right) + \frac{1}{2} \sum_{r,s=1}^T \sum_{i,j=1}^n \gamma_{ir} \gamma_{js} k_{ij} \Theta_{rs} + \lambda V(\Theta), \qquad (4)$$

where $\Theta_{rs} = \Theta(r,s)$ and $k_{ij} = k(x_i, x_j)$. Unfortunately, problem (4) is not jointly convex in $\Theta$ and $\gamma$ due to the product in the second term. A similar problem has been analyzed in [17]. They could show that for the squared loss and $V(\Theta) = \|\Theta\|_F^2$ the corresponding optimization problem is invex and directly optimize it. For an invex function every stationary point is globally optimal [22].

We follow a different path which leads to a formulation similar to the one of [2] used for learning an input mapping (see also [9]). Our formulation for the output kernel learning problem is jointly convex in the task kernel $\Theta$ and the task parameters. We present a derivation for the general RKHS $H_k$, analogous to the linear case presented in [2, 9]. We use the following variable transformation,

$$\beta_{it} = \sum_{s=1}^T \Theta_{ts} \gamma_{is}, \ i = 1, \ldots, n, \ s = 1, \ldots, T, \quad \text{resp.} \quad \gamma_{is} = \sum_{t=1}^T \left(\Theta^{-1}\right)_{st} \beta_{it}.$$

In the last expression $\Theta^{-1}$ has to be understood as the pseudo-inverse if $\Theta$ is not invertible. Note that this causes no problems as in case $\Theta$ is not invertible, we can without loss of generality restrict $\gamma$ in (4) to the range of $\Theta$. The transformation leads to our final problem formulation, where the prediction function $F$ and its squared norm $\|F\|_{H_M}^2$ can be written as

$$F(x,t) = \sum_{i=1}^n \beta_{it} k(x_i, x), \qquad \|F\|_{H_M}^2 = \sum_{r,s=1}^T \sum_{i,j=1}^n \left(\Theta^{-1}\right)_{sr} \beta_{is} \beta_{jr} k(x_i, x_j). \qquad (5)$$

We get our final primal optimization problem

$$\min_{\Theta \in S_+^T, \beta \in \mathbb{R}^{n \times T}} C \sum_{i=1}^n L\left(y_i, \sum_{j=1}^n \beta_{jt_i} k_{ji}\right) + \frac{1}{2} \sum_{r,s=1}^T \sum_{i,j=1}^n \left(\Theta^{-1}\right)_{sr} \beta_{is} \beta_{jr} k_{ij} + \lambda V(\Theta) \qquad (6)$$

Before we analyze the convexity of this problem, we want to illustrate the connection to the formulations in [9, 17]. With the task weight vectors $w_t = \sum_{j=1}^n \beta_{jt} \psi(x_j) \in H_k$ we get predictions as $F(x,t) = \langle w_t, \psi(x) \rangle$ and one can rewrite

$$\|F\|_{H_M}^2 = \sum_{r,s=1}^T \sum_{i,j=1}^n \left(\Theta^{-1}\right)_{sr} \beta_{is} \beta_{jr} k(x_i, x_j) = \sum_{r,s=1}^T \left(\Theta^{-1}\right)_{sr} \langle w_s, w_t \rangle.$$

This identity is known for vector-valued RKHS, see [15] and references therein. When $\Theta$ is $\kappa$ times the identity matrix, then $\|F\|_{H_M}^2 = \sum_{t=1}^T \frac{\|w_t\|^2}{\kappa}$ and thus (2) is learning the tasks independently. As mentioned before the convexity of the expression of $\|F\|_{H_M}^2$ is crucial for the convexity of the full problem (6). The following result has been shown in [2] (see also [9]).

**Lemma 1** *Let $\mathcal{R}(\Theta)$ denote the range of $\Theta \in S_+^T$ and let $\Theta^\dagger$ be the pseudoinverse. The extended function $f : S_+^T \times \mathbb{R}^{n \times T} \to \mathbb{R} \cup \{\infty\}$ defined as*

$$f(\Theta, \beta) = \begin{cases} \sum_{r,s=1}^T \sum_{i,j=1}^n \left(\Theta^\dagger\right)_{sr} \beta_{is} \beta_{jr} k(x_i, x_j), & \text{if } \beta_{i\cdot} \in \mathcal{R}(\Theta), \forall\, i = 1, \ldots, n, \\ \infty & \text{else}. \end{cases},$$

*is jointly convex.*

The formulation in (6) is similar to [9, 17, 18]. [9] uses the constraint $\text{Trace}(\Theta) \leq 1$ instead of a regularizer $V(\Theta)$ enforcing low rank of the output kernel. On the other hand, [17] employs squared Frobenius norm for $V(\Theta)$ with squared loss function. [18] proposed an efficient algorithm for convex $V(\Theta)$. Instead we think that sparsity of $\Theta$ is better to avoid the emergence of spurious relations between tasks and also leads to output kernels which are easier to interpret. Thus we propose to use the following regularization functional for the output kernel $\Theta$:

$$V(\Theta) = \sum_{t,t'=1}^T |\Theta_{tt'}|^p = \|\Theta\|_p^p,$$

for $p \in [1,2]$. Several approaches [9, 17, 18] employ alternate minimization scheme, involving costly eigendecompositions of $T \times T$ matrix per iteration (as $\Theta \in S_+^T$). In the next section we show that for a certain set of values of $p$ one can derive an unconstrained dual optimization problem which thus avoids the explicit minimization over the $S_+^T$ cone. The resulting *unconstrained* dual problem can then be easily optimized by stochastic coordinate ascent. Having explicit expressions of the primal variables $\Theta$ and $\beta$ in terms of the dual variables allows us to get back to the original problem.

## 3 Unconstrained Dual Problem Avoiding Optimization over $S_+^T$

The primal formulation (6) is a convex multi-task output kernel learning problem. The next lemma derives the Fenchel dual function of (6). This still involves the optimization over the primal variable $\Theta \in S_+^T$. A main contribution of this paper is to show that this optimization problem over the $S_+^T$ cone can be solved with an analytical solution for a certain class of regularizers $V(\Theta)$. In the following we denote by $\alpha^r := \{\alpha_i \mid t_i = r\}$ the dual variables corresponding to task $r$ and by $K_{rs}$ the kernel matrix $(k(x_i, x_j) \mid t_i = r, t_j = s)$ corresponding to the dual variables of tasks $r$ and $s$.

**Lemma 2** *Let $L_i^*$ be the conjugate function of the loss $L_i : \mathbb{R} \to \mathbb{R}, u \mapsto L(y_i, u)$, then*

$$q : \mathbb{R}^n \to \mathbb{R}, q(\alpha) = -C \sum_{i=1}^n L_i^*\Big(-\frac{\alpha_i}{C}\Big) - \lambda \max_{\Theta \in S_+^T} \Big(\frac{1}{2\lambda} \sum_{r,s=1}^T \Theta_{rs} \langle \alpha^r, K_{rs}\alpha^s \rangle - V(\Theta)\Big) \quad (7)$$

*is the dual function of (6), where $\alpha \in \mathbb{R}^n$ are the dual variables. The primal variable $\beta \in \mathbb{R}^{n \times T}$ in (6) and the prediction function $F$ can be expressed in terms of $\Theta$ and $\alpha$ as $\beta_{is} = \alpha_i \Theta_{st_i}$ and $F(x,s) = \sum_{j=1}^n \alpha_j \Theta_{st_j} k(x_j, x)$ respectively, where $t_j$ is the task of the $j$-th training example.*

We now focus on the remaining maximization problem in the dual function in (7)

$$\max_{\Theta \in S_+^T} \frac{1}{2\lambda} \sum_{r,s=1}^T \Theta_{rs} \langle \alpha^r, K_{rs}\alpha^s \rangle - V(\Theta). \quad (8)$$

This is a semidefinite program which is computationally expensive to solve and thus prohibits to scale the output kernel learning problem to a large number of tasks. However, we show in the following that this problem has an analytical solution for a subset of the regularizers $V(\Theta) = \frac{1}{2} \sum_{r,s=1}^T |\Theta_{rs}|^p$ for $p \geq 1$. For better readability we defer a more general result towards the end of the section. The basic idea is to relax the constraint on $\Theta \in R^{T \times T}$ in (8) so that it is equivalent to the computation of the conjugate $V^*$ of $V$. If the maximizer of the relaxed problem is positive semi-definite, one has found the solution of the original problem.

**Theorem 3** *Let $k \in \mathbb{N}$ and $p = \frac{2k}{2k-1}$, then with $\rho_{rs} = \frac{1}{2\lambda} \langle \alpha^r, K_{rs}\alpha^s \rangle$ we have*

$$\max_{\Theta \in S_+^T} \sum_{r,s=1}^T \Theta_{rs}\rho_{rs} - \frac{1}{2} \sum_{r,s=1}^T |\Theta_{rs}|^p = \frac{1}{4k-2}\Big(\frac{2k-1}{2k\lambda}\Big)^{2k} \sum_{r,s=1}^T \langle \alpha^r, K_{rs}\alpha^s \rangle^{2k}, \quad (9)$$

*and the maximizer is given by the positive semi-definite matrix*

$$\Theta_{rs}^* = \Big(\frac{2k-1}{2k\lambda}\Big)^{2k-1} \langle \alpha^r, K_{rs}\alpha^s \rangle^{2k-1}, \quad r,s = 1,\dots,T. \quad (10)$$

Plugging the result of the previous theorem into the dual function of Lemma 2 we get for $k \in \mathbb{N}$ and $p = \frac{2k}{2k-1}$ with $V(\Theta) = \|\Theta\|_p^p$ the following unconstrained dual of our main problem (6):

$$\max_{\alpha \in \mathbb{R}^n} -C \sum_{i=1}^n L_i^*\Big(-\frac{\alpha_i}{C}\Big) - \frac{\lambda}{4k-2}\Big(\frac{2k-1}{2k\lambda}\Big)^{2k} \sum_{r,s=1}^T \langle \alpha^r, K_{rs}\alpha^s \rangle^{2k}. \quad (11)$$

Note that by doing the variable transformation $\kappa_i := \frac{\alpha_i}{C}$ we effectively have only one hyper-parameter in (11). This allows us to cross-validate more efficiently. The range of admissible values for $p$ in Theorem 3 lies in the interval $(1,2]$, where we get for $k = 1$ the value $p = 2$ and as $k \to \infty$

Table 1: Examples of regularizers $V(\Theta)$ together with their generating function $\phi$ and the explicit form of $\Theta^*$ in terms of the dual variables, $\rho_{rs} = \frac{1}{2\lambda} \langle \alpha^r, K_{rs}\alpha^s \rangle$. The optimal value of (8) is given in terms of $\phi$ as $\max_{\Theta \in \mathbb{R}^{T \times T}} \langle \rho, \Theta \rangle - V(\Theta) = \sum_{r,s=1}^{T} \phi(\rho_{rs})$.

| $\phi(z)$ | $V(\Theta)$ | $\Theta^*_{rs}$ |
|---|---|---|
| $\frac{z^{2k}}{2k}$, $k \in \mathbb{N}$ | $\frac{2k-1}{2k} \sum_{r,s=1}^{T} \|\Theta_{rs}\|^{\frac{2k}{2k-1}}$ | $\rho_{rs}^{2k-1}$ |
| $e^z = \sum_{k=0}^{\infty} \frac{z^k}{k!}$ | $\begin{cases} \sum_{r,s=1}^{T} \Theta_{rs} \log(\Theta_{rs}) - \Theta_{rs} & \text{if } \Theta_{rs} > 0 \forall r,s \\ \infty & \text{else .} \end{cases}$ | $e^{\rho_{rs}}$ |
| $\cosh(z) - 1 = \sum_{k=1}^{\infty} \frac{z^{2k}}{(2k)!}$ | $\sum_{r,s=1}^{T} \left( \Theta_{rs} \operatorname{arcsinh}(\Theta_{rs}) - \sqrt{1+\Theta_{rs}^2} \right) + T^2$ | $\operatorname{arcsinh}(\rho_{rs})$ |

we have $p \to 1$. The regularizer for $p = 2$ together with the squared loss has been considered in the primal in [17, 18]. Our analytical expression of the dual is novel and allows us to employ stochastic dual coordinate ascent to solve the involved primal optimization problem. Please also note that by optimizing the dual, we have access to the duality gap and thus a well-defined stopping criterion. This is in contrast to the alternating scheme of [17, 18] for the primal problem which involves costly matrix operations. Our runtime experiments show that our solver for (11) outperforms the solvers of [17, 18]. Finally, note that even for suboptimal dual variables $\alpha$, the corresponding $\Theta$ matrix in (10) is positive semidefinite. Thus we always get a feasible set of primal variables.

**Characterizing the set of convex regularizers $V$ which allow an analytic expression for the dual function** The previous theorem raises the question for which class of convex, separable regularizers we can get an analytical expression of the dual function by explicitly solving the optimization problem (8) over the positive semidefinite cone. A key element in the proof of the previous theorem is the characterization of functions $f : \mathbb{R} \to \mathbb{R}$ which when applied elementwise $f(A) = (f(a_{ij}))_{i,j=1}^{T}$ to a positive semidefinite matrix $A \in S_+^T$ result in a p.s.d. matrix, that is $f(A) \in S_+^T$. This set of functions has been characterized by Hiai [23].

**Theorem 4 ([23])** *Let $f : \mathbb{R} \to \mathbb{R}$ and $A \in S_+^T$. We denote by $f(A) = (f(a_{ij}))_{i,j=1}^{T}$ the elementwise application of $f$ to $A$. It holds $\forall T \geq 2$, $A \in S_+^T \implies f(A) \in S_+^T$ if and only if $f$ is analytic and $f(x) = \sum_{k=0}^{\infty} a_k x^k$ with $a_k \geq 0$ for all $k \geq 0$.*

Note that in the previous theorem the condition on $f$ is only necessary when we require the implication to hold for all $T$. If $T$ is fixed, the set of functions is larger and includes even (large) fractional powers, see [24]. We use the stronger formulation as we want that the result holds without any restriction on the number of tasks $T$. Theorem 4 is the key element used in our following characterization of separable regularizers of $\Theta$ which allow an analytical expression of the dual function.

**Theorem 5** *Let $\phi : \mathbb{R} \to \mathbb{R}$ be analytic on $\mathbb{R}$ and given as $\phi(z) = \sum_{k=0}^{\infty} \frac{a_k}{k+1} z^{k+1}$ where $a_k \geq 0 \ \forall k \geq 0$. If $\phi$ is convex, then, $V(\Theta) := \sum_{r,s=1}^{T} \phi^*(\Theta_{rs})$, is a convex function $V : \mathbb{R}^{T \times T} \to \mathbb{R}$ and*

$$\max_{\Theta \in \mathbb{R}^{T \times T}} \langle \rho, \Theta \rangle - V(\Theta) = V^*(\rho) = \sum_{r,s=1}^{T} \phi(\rho_{rs}), \tag{12}$$

*where the global maximizer fulfills $\Theta^* \in S_+^T$ if $\rho \in S_+^T$ and $\Theta_{rs}^* = \sum_{k=0}^{\infty} a_k \rho_{rs}^k$.*

Table 1 summarizes e.g. of functions $\phi$, the corresponding $V(\Theta)$ and the maximizer $\Theta^*$ in (12).

## 4 Optimization Algorithm

The dual problem (11) can be efficiently solved via decomposition based methods like stochastic dual coordinate ascent algorithm (SDCA) [19]. SDCA enjoys low computational complexity per iteration and has been shown to scale effortlessly to large scale optimization problems.

**Algorithm 1** Fast MTL-SDCA
___
**Input:** Gram matrix $K$, label vector $y$, regularization parameter and relative duality gap parameter $\epsilon$
**Output:** $\alpha$ ($\Theta$ is computed from $\alpha$ using our result in 10)
Initialize $\alpha = \alpha^{(0)}$
**repeat**
    Randomly choose a dual variable $\alpha_i$
    Solve for $\Delta$ in (13) corresponding to $\alpha_i$
    $\alpha_i \leftarrow \alpha_i + \Delta$
**until** Relative duality gap is below $\epsilon$
___

Our algorithm for learning the output kernel matrix and task parameters is summarized in Algorithm 1 (refer to the supplementary material for more details). At each step of the iteration we optimize the dual objective over a randomly chosen $\alpha_i$ variable. Let $t_i = r$ be the task corresponding to $\alpha_i$. We apply the update $\alpha_i \leftarrow \alpha_i + \Delta$. The optimization problem of solving (11) with respect to $\Delta$ is as follows:

$$\min_{\Delta \in \mathbb{R}} L_i^* \big( (-\alpha_i - \Delta)/C \big) + \eta \big( (a\Delta^2 + 2b_{rr}\Delta + c_{rr})^{2k} + 2 \sum_{s \neq r} (b_{rs}\Delta + c_{rs})^{2k} + \sum_{s,z \neq r} c_{sz}^{2k} \big), \quad (13)$$

where $a = k_{ii}$, $b_{rs} = \sum_{j:t_j=s} k_{ij}\alpha_j \ \forall s$, $c_{sz} = \langle \alpha^s, K_{sz}\alpha^z \rangle \ \forall s,z$ and $\eta = \frac{\lambda}{C(4k-2)} \left( \frac{2k-1}{2k\lambda} \right)^{2k}$. This one-dimensional convex optimization problem is solved efficiently via Newton method. The complexity of the proposed algorithm is $O(T)$ per iteration . The proposed algorithm can also be employed for learning output kernels regularized by generic $V(\Theta)$, discussed in the previous section.

**Special case** $p = 2 (k = 1)$: For certain loss functions such as the hinge loss, the squared loss, etc., $L_{ti}^* \big( -\frac{\alpha_{ti}+\Delta}{C} \big)$ yields a linear or a quadratic expression in $\Delta$. In such cases problem (13) reduces to finding the roots of a cubic equation, which has a closed form expression. Hence, our algorithm is highly efficient with the above loss functions when $\Theta$ is regularized by the squared Frobenius norm.

## 5 Empirical Results

In this section, we present our results on benchmark data sets comparing our algorithm with existing approaches in terms of generalization accuracy as well as computational efficiency. Please refer to the supplementary material for additional results and details.

### 5.1 Multi-Task Data Sets

We begin with the generalization results in multi-task setups. The data sets are as follows: a) **Sarcos**: a regression data set, aim is to predict 7 degrees of freedom of a robotic arm, b) **Parkinson:** a regression data set, aim is to predict the Parkinson's disease symptom score for 42 patients, c) **Yale:** a face recognition data with 28 binary classification tasks, d) **Landmine:** a data set containing binary classifications from 19 different landmines, e) **MHC-I:** a bioinformatics data set having 10 binary classification tasks, f) **Letter:** a handwritten letters data set with 9 binary classification tasks.
We compare the following algorithms: Single task learning (STL), multi-task methods learning the output kernel matrix (MTL [16], CMTL [12], MTRL [9]) and approaches that learn both input and output kernel matrices (MTFL [11], GMTL [10]). Our proposed formulation (11) is denoted by FMTL$_p$. We consider three different values for the $p$-norm: $p = 2$ ($k = 1$), $p = 4/3$ ($k = 2$) and $p = 8/7$ ($k = 4$). Hinge and $\epsilon$-SVR loss functions were employed for classification and regression problems respectively. We follow the experimental protocol[1] described in [11].

Table 2 reports the performance of the algorithms averaged over ten random train-test splits. The proposed FMTL$_p$ attains the best generalization accuracy in general. It outperforms the baseline MTL as well as MTRL and CMTL, which solely learns the output kernel matrix. Moreover, it achieves an overall better performance than GMTL and MTFL. The FMTL$_{p=4/3,8/7}$ give comparable generalization to $p = 2$ case, with the additional benefit of learning sparser and more interpretable output kernel matrix (see Figure 1).

___
[1]The performance of STL, MTL, CMTL and MTFL are reported from [11].

Table 2: Mean generalization performance and the standard deviation over ten train-test splits.

| Data set | STL | MTL | CMTL | MTFL | GMTL | MTRL | FMTL$_p$ $p = 2$ | $p = 4/3$ | $p = 8/7$ |
|---|---|---|---|---|---|---|---|---|---|
| Regression data sets: Explained Variance (%) | | | | | | | | | |
| Sarcos | 40.5±7.6 | 34.5±10.2 | 33.0±13.4 | 49.9±6.3 | 45.8±10.6 | 41.6±7.1 | 46.7±6.9 | **50.3±5.8** | 48.4±5.8 |
| Parkinson | 2.8±7.5 | 4.9±20.0 | 2.7±3.6 | 16.8±10.8 | **33.6±9.4** | 12.0±6.8 | 27.0±4.4 | 27.0±4.4 | 27.0±4.4 |
| Classification data sets: AUC (%) | | | | | | | | | |
| Yale | 93.4±2.3 | 96.4±1.6 | 95.2±2.1 | **97.0±1.6** | 91.9±3.2 | 96.1±2.1 | **97.0±1.2** | **97.0±1.4** | 96.8±1.4 |
| Landmine | 74.6±1.6 | 76.4±0.8 | 75.9±0.7 | 76.4±1.0 | 76.7±1.2 | 76.1±1.0 | **76.8±0.8** | 76.7±1.0 | 76.4±0.9 |
| MHC-I | 69.3±2.1 | 72.3±1.9 | **72.6±1.4** | 71.7±2.2 | 72.5±2.7 | 71.5±1.7 | 71.7±1.9 | 70.8±2.1 | 70.7±1.9 |
| Letter | 61.2±0.8 | 61.0±1.6 | 60.5±1.1 | 60.5±1.8 | 61.2±0.9 | 60.3±1.4 | 61.4±0.7 | **61.5±1.0** | 61.4±1.0 |

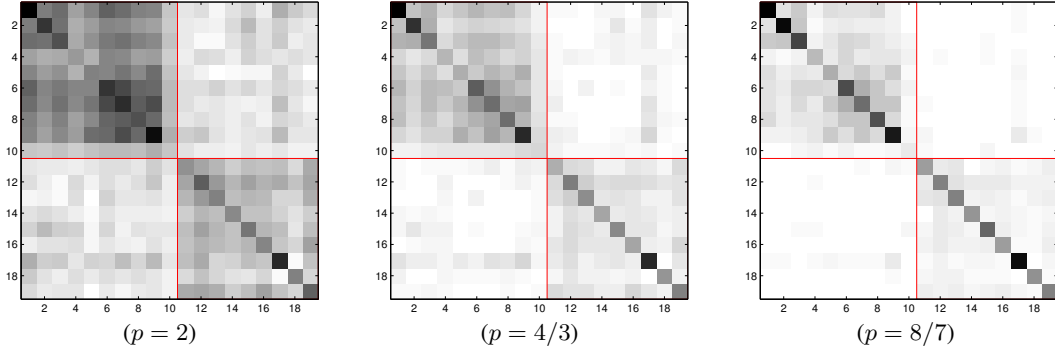

$(p = 2)$ $\qquad\qquad$ $(p = 4/3)$ $\qquad\qquad$ $(p = 8/7)$

Figure 1: Plots of $|\Theta|$ matrices (rescaled to [0,1] and averaged over ten splits) computed by our solver FMTL$_p$ for the Landmine data set for different $p$-norms, with cross-validated hyper-parameter values. The darker regions indicate higher value. Tasks (landmines) numbered 1-10 correspond to highly foliated regions and those numbered 11-19 correspond to bare earth or desert regions. Hence, we expect two groups of tasks (indicated by the red squares). We can observe that the learned $\Theta$ matrix at $p = 2$ depicts much more spurious task relationships than the ones at $p = 4/3$ and $p = 8/7$. Thus, our sparsifying regularizer improves interpretability.

Table 3: Mean accuracy and the standard deviation over five train-test splits.

| Data set | STL | MTL-SDCA | GMTL | MTRL | FMTL$_p$-H $p = 2$ | $p = 4/3$ | $p = 8/7$ | FMTL$_p$-S $p = 2$ | $p = 4/3$ | $p = 8/7$ |
|---|---|---|---|---|---|---|---|---|---|---|
| MNIST | 84.1±0.3 | 86.0±0.2 | 84.8±0.3 | 85.6±0.4 | 86.1±0.4 | 85.8±0.4 | **86.2±0.4** | 82.2±0.6 | 82.5±0.4 | 82.4±0.3 |
| USPS | 90.5±0.3 | 90.6±0.2 | 91.6±0.3 | 92.4±0.2 | 92.4±0.2 | **92.6±0.2** | **92.6±0.1** | 87.2±0.4 | 87.7±0.3 | 87.5±0.3 |

## 5.2 Multi-Class Data Sets

The multi-class setup is cast as $T$ one-vs-all binary classification tasks, corresponding to $T$ classes. In this section we experimented with two loss functions: a) FMTL$_p$-H – the hinge loss employed in SVMs, and b) FMTL$_p$-S – the squared loss employed in OKL [17]. In these experiments, we also compare our results with **MTL-SDCA**, a state-of-the-art multi-task feature learning method [25].

**USPS & MNIST Experiments**: We followed the experimental protocol detailed in [10]. Results are tabulated in Table 3. Our approach FMTL$_p$-H obtains better accuracy than GMTL, MTRL and MTL-SDCA [25] on both data sets.

**MIT Indoor67 Experiments**: We report results on the MIT Indoor67 benchmark [26] which covers 67 indoor scene categories. We use the train/test split (80/20 images per class) provided by the authors. FMTL$_p$-S achieved the accuracy of 73.3% with $p = 8/7$. Note that this is better than the ones reported in [27] (70.1%) and [26] (68.24%).

**SUN397 Experiments**: SUN397 [28] is a challenging scene classification benchmark [26] with 397 classes. We use $m = 5, 50$ images per class for training, 50 images per class for testing and report the average accuracy over the 10 standard splits. We employed the CNN features extracted with the

Table 4: Mean accuracy and the standard deviation over ten train-test splits on SUN397.

| $m$ | STL | MTL | MTL-SDCA | FMTL$_p$-H | | | FMTL$_p$-S | | |
|---|---|---|---|---|---|---|---|---|---|
| | | | | $p = 2$ | $p = 4/3$ | $p = 8/7$ | $p = 2$ | $p = 4/3$ | $p = 8/7$ |
| 5 | 40.5±0.9 | 42.0±1.4 | 41.2±1.3 | 41.5±1.1 | 41.6±1.3 | 41.6±1.2 | **44.1±1.3** | **44.1±1.1** | 44.0±1.2 |
| 50 | 55.0±0.4 | 57.0±0.2 | 54.8±0.3 | 55.1±0.2 | 55.6±0.3 | 55.1±0.3 | **58.6±0.1** | 58.5±0.1 | **58.6±0.2** |

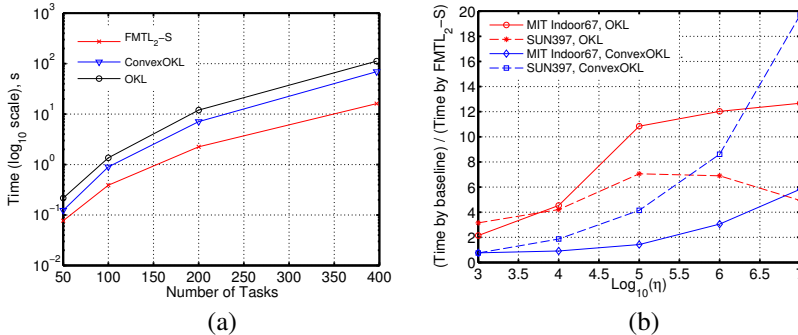

(a)                    (b)

Figure 2: (a) Plot compares the runtime of various algorithms with varying number of tasks on SUN397. Our approach FMTL$_2$-S is 7 times faster that OKL [17] and 4.3 times faster than Convex-OKL [18] when the number of tasks is maximum. (b) Plot showing the factor by which FMTL$_2$-S outperforms OKL and ConvexOKL over the hyper-parameter range on various data sets. On SUN397, we outperform OKL and ConvexOKL by factors of 5.2 and 7 respectively. On MIT Indoor67, we are better than OKL and ConvexOKL by factors of 8.4 and 2.4 respectively.

convolutional neural network (CNN) [26] using Places 205 database. The results are tabulated in Table 4. The $\Theta$ matrices computed by FMTL$_p$-S are discussed in the supplementary material.

## 5.3  Scaling Experiment

We compare the runtime of our solver for FMTL$_2$-S with the OKL solver of [17] and the Convex-OKL solver of [18] on several data sets. All the three methods solve the same optimization problem. Figure 2a shows the result of the scaling experiment where we vary the number of tasks (classes). The parameters employed are the ones obtained via cross-validation. Note that both OKL and Convex-OKL algorithms do not have a well defined stopping criterion whereas our approach can easily compute the relative duality gap (set as $10^{-3}$). We terminate them when they reach the primal objective value achieved by FMTL$_2$-S . Our optimization approach is 7 times and 4.3 times faster than the alternate minimization based OKL and ConvexOKL, respectively, when the number of tasks is maximal. The generic FMTL$_{p=4/3,8/7}$ are also considerably faster than OKL and ConvexOKL.

Figure 2b compares the average runtime of our FMTL$_p$-S with OKL and ConvexOKL on the cross-validated range of hyper-parameter values. FMTL$_p$-S outperform them on both MIT Indoor67 and SUN397 data sets. On MNIST and USPS data sets, FMTL$_p$-S is more than 25 times faster than OKL, and more than 6 times faster than ConvexOKL. Additional details of the above experiments are discussed in the supplementary material.

## 6  Conclusion

We proposed a novel formulation for learning the positive semi-definite output kernel matrix for multiple tasks. Our main technical contribution is our analysis of a certain class of regularizers on the output kernel matrix where one may drop the positive semi-definite constraint from the optimization problem, but still solve the problem optimally. This leads to a dual formulation that can be efficiently solved using stochastic dual coordinate ascent algorithm. Results on benchmark multi-task and multi-class data sets demonstrates the effectiveness of the proposed multi-task algorithm in terms of runtime as well as generalization accuracy.

**Acknowledgments.** P.J. and M.H. acknowledge the support by the Cluster of Excellence (MMCI).

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
