[Supplementary Material]

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

$. In order to see this we first need the following representer theorem for fixed output kernel $\Theta$.

**Lemma 1** *The optimal solution $F^* \in H_M$ of the optimization problem*

$$\min_{F \in H_M} C \sum_{i=1}^{n} L(y_i, F(x_i, t_i)) + \frac{1}{2} \|F\|_{H_M}^2 \tag{3}$$

*admits a representation of the form*

$$F^*(x, t) = \sum_{s=1}^{T} \sum_{i=1}^{n} \gamma_{is} M((x_i, s), (x, t)) = \sum_{s=1}^{T} \sum_{i=1}^{n} \gamma_{is} k(x_i, x) \Theta(s, t),$$

*where $F^*(x, t)$ is the prediction for instance $x$ belonging to task $t$ and $\gamma \in \mathbb{R}^{n \times T}$.*

**Proof:** The proof is analogous to the standard representer theorem [20]. We denote by $U = \text{Span}(M((x_i, s), (\cdot, \cdot)) \,|\, i = 1, \ldots, n, \, s = 1, \ldots, T)$ the subspace in $H_M$ spanned by the training data. This induces the orthogonal decomposition of $H_M = U \oplus U^\perp$, where $U^\perp$ is the orthogonal subpace of $U$. Every function $F \in H_M$ can correspondingly decomposed into $F = F^\parallel + F^\perp$, where $F^\parallel \in U$ and $F^\perp \in U^\perp$. Then $\|F\|_{H_M}^2 = \|F^\parallel\|_{H_M}^2 + \|F^\perp\|_{H_M}^2$. As

$$F(x_i, t_i) = \langle F, M((x_i, t_i), (\cdot, \cdot)) \rangle = \langle F^\parallel, M((x_i, t_i), (\cdot, \cdot)) \rangle = F^\parallel(x_i, t_i). \tag{4}$$

As the loss only depends on $F^\parallel$ and we minimize the objective by having $\|F^\perp\|_{H_M} = 0$. This yields the result. $\qquad\square$

With the explicit form of the prediction function one can rewrite the main problem (2) as

$$\min_{\Theta \in S_+^T, \gamma \in \mathbb{R}^{n \times T}} C \sum_{i=1}^{n} L\left(y_i, \sum_{s=1}^{T} \sum_{j=1}^{n} \gamma_{js} k_{ji} \Theta_{s\,t_i}\right) + \frac{1}{2} \sum_{r,s=1}^{T} \sum_{i,j=1}^{n} \gamma_{ir} \gamma_{js} k_{ij} \Theta_{rs} + \lambda V(\Theta), \tag{5}$$

where $\Theta_{rs} = \Theta(r, s)$ and $k_{ij} = k(x_i, x_j)$. Unfortunately, problem (5) is not jointly convex in $\Theta$ and $\gamma$ due to the product in the second term. A similar problem has been analyzed in [17]. They could show that for the squared loss and $V(\Theta) = \|\Theta\|_F^2$ the corresponding optimization problem is invex and directly optimize it. For an invex function every stationary point is globally optimal [22].

We follow a different path which leads to a formulation similar to the one of [2] used for learning an input mapping (see also [9]). Our formulation for the output kernel learning problem is jointly convex in the task kernel $\Theta$ and the task parameters. We present a derivation for the general RKHS $H_k$, analogous to the linear case presented in [2, 9]. We use the following variable transformation,

$$\beta_{it} = \sum_{s=1}^{T} \Theta_{ts} \gamma_{is}, \; i = 1, \ldots, n, \; s = 1, \ldots, T, \quad \text{resp.} \quad \gamma_{is} = \sum_{t=1}^{T} \left(\Theta^{-1}\right)_{st} \beta_{it}.$$

In the last expression $\Theta^{-1}$ has to be understood as the pseudo-inverse if $\Theta$ is not invertible. Note that this causes no problems as in case $\Theta$ is not invertible, we can without loss of generality restrict $\gamma$ in (5) to the range of $\Theta$. The transformation leads to our final problem formulation, where the prediction function $F$ and its squared norm $\|F\|_{H_M}^2$ can be written as

$$F(x, t) = \sum_{i=1}^{n} \beta_{it} k(x_i, x), \qquad \|F\|_{H_M}^2 = \sum_{r,s=1}^{T} \sum_{i,j=1}^{n} \left(\Theta^{-1}\right)_{sr} \beta_{is} \beta_{jr} k(x_i, x_j). \tag{6}$$

This can be seen as follows

$$\|F\|_{H_M}^2 = \sum_{r,s=1}^{T} \sum_{i,j=1}^{n} \gamma_{ir} \gamma_{js} k(x_i, x_j) \Theta_{rs} \tag{7}$$

$$= \sum_{t,u=1}^{T} \sum_{r,s=1}^{T} \sum_{i,j=1}^{n} \beta_{it} \beta_{ju} \left(\Theta^{-1}\right)_{tr} \left(\Theta^{-1}\right)_{us} k(x_i, x_j) \Theta_{rs} \tag{8}$$

$$= \sum_{t,u=1}^{T} \sum_{i,j=1}^{n} \left(\Theta^{-1}\right)_{tu} \beta_{it} \beta_{ju} k(x_i, x_j). \tag{9}$$

We get our final primal optimization problem

$$\min_{\Theta \in S_+^T, \beta \in \mathbb{R}^{n \times T}} C \sum_{i=1}^n L\left(y_i, \sum_{j=1}^n \beta_{jt_i} k_{ji}\right) + \frac{1}{2} \sum_{r,s=1}^T \sum_{i,j=1}^n \left(\Theta^{-1}\right)_{sr} \beta_{is} \beta_{jr} k_{ij} + \lambda V(\Theta) \qquad (10)$$

Before we analyze the convexity of this problem, we want to illustrate the connection to the formulations in [9, 17]. With the task weight vectors $w_t = \sum_{j=1}^n \beta_{jt} \psi(x_j) \in H_k$ we get predictions as $F(x,t) = \langle w_t, \psi(x) \rangle$ and one can rewrite

$$\|F\|_{H_M}^2 = \sum_{r,s=1}^T \sum_{i,j=1}^n \left(\Theta^{-1}\right)_{sr} \beta_{is} \beta_{jr} k(x_i, x_j) = \sum_{r,s=1}^T \left(\Theta^{-1}\right)_{sr} \langle w_s, w_t \rangle .$$

This identity is known for vector-valued RKHS, see [15] and references therein. When $\Theta$ is $\kappa$ times the identity matrix, then $\|F\|_{H_M}^2 = \sum_{t=1}^T \frac{\|w_t\|^2}{\kappa}$ and thus (2) is learning the tasks independently. As mentioned before the convexity of the expression of $\|F\|_{H_M}^2$ is crucial for the convexity of the full problem (10). The following result has been shown in [2] (see also [9]).

**Lemma 2** *Let $\mathcal{R}(\Theta)$ denote the range of $\Theta \in S_+^T$ and let $\Theta^\dagger$ be the pseudoinverse. The extended function $f : S_+^T \times \mathbb{R}^{n \times T} \to \mathbb{R} \cup \{\infty\}$ defined as*

$$f(\Theta, \beta) = \begin{cases} \sum_{r,s=1}^T \sum_{i,j=1}^n \left(\Theta^\dagger\right)_{sr} \beta_{is} \beta_{jr} k(x_i, x_j), & \text{if } \beta_{i\cdot} \in \mathcal{R}(\Theta), \forall\, i = 1, \ldots, n, \\ \infty & \text{else} . \end{cases}$$

*is jointly convex.*

**Proof:** It has been shown in [2] and [23][p. 223] that $\langle x, A^\dagger x \rangle$ is jointly convex on $S_+^T \times \mathcal{R}(A)$, where $\mathcal{R}(A)$ is the range of $A$ and $A^\dagger$ is the pseudoinverse of $A \in S_+^T$. As $L := (k(x_i, x_j))_{i,j=1}^n$ is positive semi-definite we can compute the eigendecomposition as

$$L_{ij} = \sum_{l=1}^n \lambda_l u_{li} u_{lj},$$

where $\lambda_l \geq 0$, $l = 1, \ldots, n$ are the eigenvalues and $u_l \in \mathbb{R}^n$ the eigenvectors. Using this we get

$$\sum_{r,s=1}^T \sum_{i,j=1}^n \left(\Theta^{-1}\right)_{sr} \beta_{is} \beta_{jr} k(x_i, x_j) = \sum_{l=1}^n \lambda_l \sum_{r,s=1}^T \left(\sum_{i=1}^n \beta_{is} u_{li}\right)\left(\sum_{j=1}^n \beta_{jr} u_{lj}\right)\left(\Theta^{-1}\right)_{rs} \qquad (11)$$

and thus we can write the function $f$ as a positive combination of convex functions, where the arguments are composed with linear mappings which preserves convexity [24]. □

The formulation in (10) is similar to [9, 17, 18]. [9] uses the constraint $\text{Trace}(\Theta) \leq 1$ instead of a regularizer $V(\Theta)$ enforcing low rank of the output kernel. On the other hand, [17] employs squared Frobenius norm for $V(\Theta)$ with squared loss function. [18] proposed an efficient algorithm for convex $V(\Theta)$. Instead we think that sparsity of $\Theta$ is better to avoid the emergence of spurious relations between tasks and also leads to output kernels which are easier to interpret. Thus we propose to use the following regularization functional for the output kernel $\Theta$:

$$V(\Theta) = \sum_{t,t'=1}^T |\Theta_{tt'}|^p = \|\Theta\|_p^p,$$

for $p \in [1, 2]$. Several approaches [9, 17, 18] employ alternate minimization scheme, involving costly eigendecompositions of $T \times T$ matrix per iteration (as $\Theta \in S_+^T$). In the next section we show that for a certain set of values of $p$ one can derive an unconstrained dual optimization problem which thus avoids the explicit minimization over the $S_+^T$ cone. The resulting *unconstrained* dual problem can then be easily optimized by stochastic coordinate ascent. Having explicit expressions of the primal variables $\Theta$ and $\beta$ in terms of the dual variables allows us to get back to the original problem.

# 3 Unconstrained Dual Problem Avoiding Optimization over $S_+^T$

The primal formulation (10) is a convex multi-task output kernel learning problem. The next lemma derives the Fenchel dual function of (10). This still involves the optimization over the primal variable $\Theta \in S_+^T$. A main contribution of this paper is to show that this optimization problem over the $S_+^T$ cone can be solved with an analytical solution for a certain class of regularizers $V(\Theta)$. In the following we denote by $\alpha^r := \{\alpha_i \mid t_i = r\}$ the dual variables corresponding to task $r$ and by $K_{rs}$ the kernel matrix $(k(x_i, x_j) \mid t_i = r, t_j = s)$ corresponding to the dual variables of tasks $r$ and $s$.

**Lemma 3** *Let $L_i^*$ be the conjugate function of the loss $L_i : \mathbb{R} \to \mathbb{R}, u \mapsto L(y_i, u)$, then*

$$q : \mathbb{R}^n \to \mathbb{R}, q(\alpha) = -C \sum_{i=1}^n L_i^* \Big( -\frac{\alpha_i}{C} \Big) - \lambda \max_{\Theta \in S_+^T} \Big( \frac{1}{2\lambda} \sum_{r,s=1}^T \Theta_{rs} \langle \alpha^r, K_{rs} \alpha^s \rangle - V(\Theta) \Big) \quad (12)$$

*is the dual function of (10), where $\alpha \in \mathbb{R}^n$ are the dual variables. The primal variable $\beta \in \mathbb{R}^{n \times T}$ in (10) and the prediction function $F$ can be expressed in terms of $\Theta$ and $\alpha$ as $\beta_{is} = \alpha_i \Theta_{st_i}$ and $F(x, s) = \sum_{j=1}^n \alpha_j \Theta_{st_j} k(x_j, x)$ respectively, where $t_j$ is the task of the $j$-th training example.*

**Proof:** We derive the Fenchel dual function of (10). For this purpose we introduce auxiliary variables $z \in \mathbb{R}^n$ which satisfy the constraint

$$z_i = \sum_{j=1}^n \beta_{jt_i} k(x_j, x_i) = F(x_i, t_i).$$

The Lagrangian $L$ of the resulting problem (10) is given as:

$$L(\beta, \Theta, z, \alpha) = C \sum_{i=1}^n L(y_i, z_i) + \frac{1}{2} \sum_{r,s=1}^T \sum_{i,j=1}^n \big(\Theta^{-1}\big)_{sr} \beta_{is} \beta_{jr} k(x_i, x_j) \quad (13)$$

$$+ \sum_{i=1}^n \alpha_i \Big( z_i - \sum_{j=1}^n \beta_{jt_i} k(x_j, x_i) \Big) + i_{S_+^T}(\Theta) + \lambda V(\Theta).$$

where $i_C$ is the indicator function of the set $C$. The dual function $q$ is defined as

$$q(\alpha) = \min_{\beta \in \mathbb{R}^{n \times T}, \Theta \in S_+^T, z \in \mathbb{R}^n} L(\beta, \Theta, z, \alpha). \quad (14)$$

Using the definition of the conjugate function [24], we get

$$\min_{z_i \in \mathbb{R}} C\, L(y_i, z_i) + \alpha_i z_i = C \min_{z_i \in \mathbb{R}} L(y_i, z_i) + \frac{\alpha_i}{C} z_i = -C \max_{z_i \in \mathbb{R}} \Big( -\frac{\alpha_i}{C} z_i - L(y_i, z_i) \Big) \quad (15)$$

$$= -C\, L_i^* \Big( -\frac{\alpha_i}{C} \Big), \quad (16)$$

where $L_i^*$ is the conjugate function of $L_i : z \to L(y_i, z)$. Moreover, we compute the minimizer with respect to $\beta$, via

$$\frac{\partial}{\partial \beta_{lu}} \Big( \frac{1}{2} \sum_{r,s=1}^T \sum_{i,j=1}^n \big(\Theta^{-1}\big)_{sr} \beta_{is} \beta_{jr} k(x_i, x_j) - \sum_{i=1}^n \alpha_i \Big( \sum_{j=1}^n \beta_{jt_i} k(x_j, x_i) \Big) \quad (17)$$

$$= \sum_{r=1}^T \sum_{j=1}^n \beta_{jr} (\Theta^{-1})_{ur} k(x_l, x_j) - \sum_{i=1}^n \alpha_i \delta_{ut_i} k(x_l, x_i),$$

where $\delta$ is the Kronecker symbol, that is $\delta_{ut_i} = \begin{cases} 1 & \text{if } u = t_i, \\ 0 & \text{else} \end{cases}$. Solving for the global minimizer $\beta^*$ yields

$$\beta_{jr}^* = \alpha_j \Theta_{rt_j}. \quad (18)$$

Plugging $\beta^*$ back into the above expressions yields

$$\sum_{r,s=1}^{T} \sum_{i,j=1}^{n} \left(\Theta^{-1}\right)_{sr} \beta_{is} \beta_{jr} k(x_i, x_j) = \sum_{r,s=1}^{T} \sum_{i,j=1}^{n} \left(\Theta^{-1}\right)_{sr} \Theta_{st_i} \Theta_{rt_j} \alpha_i \alpha_j k(x_i, x_j)$$

$$= \sum_{i,j=1}^{n} \Theta_{t_i t_j} \alpha_i \alpha_j k(x_i, x_j), \tag{19}$$

$$\sum_{i,j=1}^{n} \alpha_i \beta_{jt_i} k(x_j, x_i) = \sum_{i,j=1}^{n} \alpha_i \alpha_j \Theta_{t_j t_i} k(x_j, x_i), \tag{20}$$

Introducing $\alpha^r = (\alpha_i)_{t_i=r}$, $K_{rs} = \left(k(x_i, x_j)\right)_{t_i=r, t_j=s}$ and gathering the terms corresponding to the individual tasks we get

$$\sum_{i,j=1}^{n} \alpha_i \alpha_j \Theta_{t_j t_i} k(x_j, x_i) = \sum_{r,s=1}^{T} \langle \alpha^r, K_{rs} \alpha^s \rangle .$$

Plugging all the expressions back into (14), we get the dual function as

$$q(\alpha) = -C L_{ti}^* (-\frac{\alpha_{ti}}{C}) + \min_{\Theta \in S_+^T} \lambda V(\Theta) - \frac{1}{2} \sum_{r,s=1}^{T} \Theta_{rs} \langle \alpha^r, K_{rs} \alpha^s \rangle \tag{21}$$

$$= -C L_{ti}^* (-\frac{\alpha_{ti}}{C}) + \lambda \min_{\Theta \in S_+^T} V(\Theta) - \langle \rho, \Theta \rangle \tag{22}$$

$$= -C L_{ti}^* (-\frac{\alpha_{ti}}{C}) - \lambda \max_{\Theta \in S_+^T} \langle \rho, \Theta \rangle - V(\Theta) \tag{23}$$

where we have introduced in the second step $\rho \in \mathbb{R}^{T \times T}$ with

$$\rho_{rs} = \frac{1}{2\lambda} \langle \alpha^r, K_{rs} \alpha^s \rangle, \quad r,s = 1, \ldots, T.$$

Note that $\rho$ is a Gram matrix and thus positive semidefinite. The expression for the prediction function is obtained by plugging (18) into (6). $\qquad\square$

We now focus on the remaining maximization problem in the dual function in (12)

$$\max_{\Theta \in S_+^T} \frac{1}{2\lambda} \sum_{r,s=1}^{T} \Theta_{rs} \langle \alpha^r, K_{rs} \alpha^s \rangle - V(\Theta). \tag{24}$$

This is a semidefinite program which is computationally expensive to solve and thus prohibits to scale the output kernel learning problem to a large number of tasks. However, we show in the following that this problem has an analytical solution for a subset of the regularizers $V(\Theta) = \frac{1}{2} \sum_{r,s=1}^{T} |\Theta_{rs}|^p$ for $p \geq 1$. For better readability we defer a more general result towards the end of the section. The basic idea is to relax the constraint on $\Theta \in R^{T \times T}$ in (24) so that it is equivalent to the computation of the conjugate $V^*$ of $V$. If the maximizer of the relaxed problem is positive semi-definite, one has found the solution of the original problem.

**Theorem 4** *Let $k \in \mathbb{N}$ and $p = \frac{2k}{2k-1}$, then with $\rho_{rs} = \frac{1}{2\lambda} \langle \alpha^r, K_{rs} \alpha^s \rangle$ we have*

$$\max_{\Theta \in S_+^T} \sum_{r,s=1}^{T} \Theta_{rs} \rho_{rs} - \frac{1}{2} \sum_{r,s=1}^{T} |\Theta_{rs}|^p = \frac{1}{4k-2} \left(\frac{2k-1}{2k\lambda}\right)^{2k} \sum_{r,s=1}^{T} \langle \alpha^r, K_{rs} \alpha^s \rangle^{2k}, \tag{25}$$

*and the maximizer is given by the positive semi-definite matrix*

$$\Theta_{rs}^* = \left(\frac{2k-1}{2k\lambda}\right)^{2k-1} \langle \alpha^r, K_{rs} \alpha^s \rangle^{2k-1}, \quad r,s = 1, \ldots, T. \tag{26}$$

**Proof:** We relax the constraints and solve

$$\max_{\Theta \in \mathbb{R}^{T \times T}} \frac{1}{2\lambda} \sum_{r,s=1}^{T} \Theta_{rs} \langle \alpha^r, K_{rs}\alpha^s \rangle - \frac{1}{2} \sum_{r,s=1}^{T} |\Theta_{rs}|^p.$$

Note that the problem is separable and thus we can solve for each component separately,

$$\max_{\Theta_{rs} \in \mathbb{R}} \frac{1}{2\lambda} \Theta_{rs} \langle \alpha^r, K_{rs}\alpha^s \rangle - \frac{1}{2} |\Theta_{rs}|^p.$$

The optimality condition for $\Theta_{rs}^*$ becomes with $\rho_{rs} = \frac{1}{2\lambda} \langle \alpha^r, K_{rs}\alpha^s \rangle$,

$$0 = \rho_{rs} - \frac{p}{2} \operatorname{sign}(\Theta_{rs}^*)|\Theta_{rs}^*|^{p-1} \implies \Theta_{rs}^* = \left(\frac{2}{p}\right)^{\frac{1}{p-1}} \operatorname{sign}(\rho_{rs})|\rho_{rs}|^{\frac{1}{p-1}}.$$

The solution of the relaxed problem is the solution of the original constrained problem, if we can show that the corresponding maximizer is positive semidefinite. Note that $\rho_{rs} = \frac{1}{2\lambda} \langle \alpha^r, K_{rs}\alpha^s \rangle$ is a positive semidefinite (p.s.d.) matrix as it is a Gram matrix. The factor $\left(\frac{2}{p}\right)^{\frac{1}{p-1}}$ is positive and thus the resulting matrix is p.s.d. if $\operatorname{sign}(\rho_{rs})|\rho_{rs}|^{\frac{1}{p-1}}$ is p.s.d.

It has been shown [25], that the elementwise power $A_{rs}^l$ of a positive semidefinite matrix $A$ is positive definite for all $A \in S_+^T$ and $T \in \mathbb{N}$ if and only if $l$ is a positive integer. Note that we have an elementwise integer power of $\Theta$ if $\frac{1}{p-1}$ is an odd positive integer (the case of an even integer is ruled out by Theorem 5), that is $\frac{1}{p-1} = 2k - 1$ for $k \in \mathbb{N}$ as in this case we have

$$\Theta_{rs}^* = \left(\frac{2}{p}\right)^{2k-1} \operatorname{sign}(\rho_{rs})|\rho_{rs}|^{2k-1} = \left(\frac{2}{p}\right)^{2k-1} \rho_{rs}^{2k-1} = \left(\frac{2k-1}{2k\lambda}\right)^{2k-1} \langle \alpha^r, K_{rs}\alpha^s \rangle^{2k-1}.$$

We get the admissible values of $p$ as $p = \frac{2k}{2k-1}$, $k \in \mathbb{N}$ (resp. $2k = \frac{p}{p-1}$). We compute the optimal objective value as

$$\sum_{r,s=1}^{T} \rho_{rs}^{2k}\left(\left(\frac{2}{p}\right)^{2k-1} - \frac{1}{2}\left(\frac{2}{p}\right)^{2k}\right) = (p-1)\frac{1}{2}\left(\frac{2}{p}\right)^{2k} \sum_{r,s=1}^{T} \rho_{rs}^{2k} = \frac{1}{4k-2}\left(\frac{2k-1}{k}\right)^{2k} \sum_{r,s=1}^{T} \rho_{rs}^{2k} \tag{27}$$

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

**Proof:** Note that $\phi$ is analytic on $\mathbb{R}$ and thus infinitely differentiable on $\mathbb{R}$. As $\phi$ is additionally convex, it is a proper, lower semi-continuous convex function and thus $(\phi^*)^* = \phi$ [27, Corollary 1.3.6]. As $\phi^*$ is convex, $V$ is a convex function and using $(\phi^*)^* = \phi$ we get

$$\max_{\Theta \in \mathbb{R}^{T \times T}} \langle \rho, \Theta \rangle - V(\Theta) = V^*(\rho) = \sum_{r,s=1}^T \phi(\rho_{rs}). \tag{31}$$

Finally, we show that the global maximizer has the given form. Note that as $\phi$ is a proper, lower semi-continuous convex function it holds [27, Corollary 1.4.4]

$$\Theta_{rs} \in \partial \phi^*(\rho_{rs}) \quad \Longleftrightarrow \quad \rho_{rs} \in \partial \phi(\Theta_{rs}).$$

Note that the maximizer $\Theta_{rs}^*$ of problem (31) fulfills $\rho_{rs} \in \frac{\partial \phi^*}{\partial \Theta_{rs}}(\Theta_{rs}^*)$ and thus $\Theta_{rs}^* = \frac{\partial \phi}{\partial \rho_{rs}}(\rho_{rs})$, where we have used that $\phi$ is infinitely differentiable. These conditions allow us to express the maximizer of (30) in terms of $\partial \phi$. As $\phi$ is continuously differentiable, we get

$$\Theta_{rs}^* = \frac{\partial \phi}{\partial \rho_{rs}}(\rho_{rs}) = \sum_{k=0}^\infty a_k \rho_{rs}^k.$$

Note that the series has infinite convergence radius and $a_k \geq 0$ for all $k$ and thus it is of the form provided in Theorem 5. Thus $\Theta^* \in S_+^T$ if $\rho \in S_+^T$. $\qquad\square$

Table 1 summarizes e.g. of functions $\phi$, the corresponding $V(\Theta)$ and the maximizer $\Theta^*$ in (30).

**Examples**

- First we recover the results of Theorem 4. We use $\phi(x) = \frac{1}{2k}x^{2k}$ for $k \in \mathbb{N}$, which is convex. We compute

$$\phi^*(y) = \sup_{x \in \mathbb{R}} xy - \phi(x) = \sup_{x \in \mathbb{R}} xy - \frac{1}{2k}x^{2k} = \frac{2k-1}{2k}|y|^{\frac{2k}{2k-1}},$$

where we have used $x^* = |y|^{\frac{1}{2k-1}}\mathrm{sign}(y)$. We recover

$$V(\Theta) = \sum_{r,s=1}^{T} \phi^*(\Theta_{rs}) = \frac{2k-1}{2k} \sum_{r,s=1}^{T} \Theta_{rs}^{\frac{2k}{2k-1}},$$

which with $p = \frac{2k}{2k-1}$ yields up to a positive factor the family of regularizers employed in Theorem 4 together with

$$\Theta_{rs}^* = \rho_{rs}^{2k-1}$$

- In the second example we use $\phi(x) = e^x = \sum_{k=0}^{\infty} \frac{x^k}{k!}$, which is convex and the series has infinite convergence radius The conjugate $\phi^*$ is given as

$$\phi^*(y) = \sup_{x \in \mathbb{R}} xy - e^x = \begin{cases} y\log(y) - y & \text{if } y > 0 \\ \infty & \text{else.} \end{cases}$$

so that the regularizer is given by,

$$V(\Theta) = \sum_{r,s=1}^{T} \phi^*(\Theta_{rs}) = \begin{cases} \sum_{r,s=1}^{T} \Theta_{rs}\log(\Theta_{rs}) - \Theta_{rs} & \text{if } \Theta_{rs} > 0 \ \forall r, s = 1, \ldots, T \\ \infty & \text{else .} \end{cases}.$$

This can be seen as a generalized KL-divergence between $\Theta$ and $\Theta_0$, where $\Theta_0 \in S_+^T$ is the matrix of all ones

$$V(\Theta) = \sum_{r,s=1}^{T} \phi^*(\Theta_{rs}) = \begin{cases} \sum_{r,s=1}^{T} \Theta_{rs}\log\left(\frac{\Theta_{rs}}{(\Theta_0)_{rs}}\right) - \Theta_{rs} + (\Theta_0)_{rs} & \text{if } \Theta_{rs} > 0 \ \forall r, s \\ \infty & \text{else .} \end{cases}.$$

Note that adding the constant term $\sum_{r,s=1}^{T} (\Theta_0)_{rs}$ does not change the optimization problem (10). The corresponding $\Theta^*$ is given by

$$\Theta_{rs}^* = \sum_{k=0}^{\infty} \frac{\rho_{rs}^k}{k!} = e^{\rho_{rs}}.$$

- Next we consider $\phi(x) = \cosh(x) - 1 = \sum_{k=1}^{\infty} \frac{x^{2k}}{(2k)!}$ which is obviously convex and the series has infinite convergence radius ($e^x$ is majorant). The conjugate $\phi^*$ can be computed as

$$\phi^*(y) = \sup_{x \in \mathbb{R}} xy - \cosh(x) + 1 = y\,\mathrm{arcsinh}(y) - \sqrt{1 + y^2} + 1 = y\log(y + \sqrt{y^2 + 1}) - \sqrt{1 + y^2} + 1.$$

so that the regularizer is given by

$$V(\Theta) = \sum_{r,s=1}^{T} \phi^*(\Theta_{rs}) = \sum_{r,s=1}^{T} \left(\Theta_{rs}\,\mathrm{arcsinh}(\Theta_{rs}) - \sqrt{1 + \Theta_{rs}^2} + 1\right).$$

The corresponding $\Theta^*$ is given by

$$\Theta_{rs}^* = \mathrm{arcsinh}(\rho_{rs}) = \log\left(\rho_{rs} + \sqrt{\rho_{rs}^2 + 1}\right).$$

This regularizer is interpolating between a squared norm and a variant of 1-norm. One has

$$\lim_{y \to 0} \phi^*(y) = \frac{y^2}{2}, \quad \lim_{y \to \infty} \phi^*(y) = |y|(\log(2|y|) - 1) + 1.$$

---

**Algorithm 1** Fast MTL-SDCA

> **Input:** Gram matrix $K$, label vector $y$, regularization parameter and relative duality gap parameter $\epsilon$
> **Output:** $\alpha$ ($\Theta$ is computed from $\alpha$ using our result in 26)
> Initialize $\alpha = \alpha^{(0)}$
> **repeat**
>     Let $\{i_1, \ldots, i_n\}$ be a random permutation of $\{1, \ldots, n\}$
>     **for** $j = 1, \ldots, n$ **do**
>         Solve for $\Delta$ in (32) corresponding to $\alpha_{i_j}$
>         $\alpha_{i_j} \leftarrow \alpha_{i_j} + \Delta$
>     **end for**
> **until** Relative duality gap is below $\epsilon$

---

## 4 Optimization Algorithm

The dual problem (29) can be efficiently solved via decomposition based methods like stochastic dual coordinate ascent algorithm (SDCA) [19]. SDCA enjoys low computational complexity per iteration and has been shown to scale effortlessly to large scale optimization problems.

Our algorithm for learning the output kernel matrix and task parameters is summarized in Algorithm 1. At each step of the iteration we optimize the dual objective over a randomly chosen $\alpha_i$ variable. Let $t_i = r$ be the task corresponding to $\alpha_i$. We apply the update $\alpha_i \leftarrow \alpha_i + \Delta$. The optimization problem of solving (29) with respect to $\Delta$ is as follows:

$$\min_{\Delta \in \mathbb{R}} L_i^* \big( (-\alpha_i - \Delta)/C \big) + \eta \Big( (a\Delta^2 + 2b_{rr}\Delta + c_{rr})^{2k} + 2 \sum_{s \neq r} (b_{rs}\Delta + c_{rs})^{2k} + \sum_{s,z \neq r} c_{sz}^{2k} \Big), \quad (32)$$

where $a = k_{ii}$, $b_{rs} = \sum_{j:t_j=s} k_{ij}\alpha_j \ \forall s$, $c_{sz} = \langle \alpha^s, K_{sz}\alpha^z \rangle \ \forall s, z$ and $\eta = \frac{\lambda}{C(4k-2)} \left( \frac{2k-1}{2k\lambda} \right)^{2k}$.
This one-dimensional convex optimization problem is solved efficiently via Newton method. The complexity of the proposed algorithm is $O(T)$ per iteration . The proposed algorithm can also be employed for learning output kernels regularized by generic $V(\Theta)$, discussed in the previous section.

**Special case** $p = 2(k = 1)$: For certain loss functions such as the hinge loss, the squared loss, etc., $L_{ti}^* \big( -\frac{\alpha_{ti}+\Delta}{C} \big)$ yields a linear or a quadratic expression in $\Delta$. In such cases problem (32) reduces to finding the roots of a cubic equation, which has a closed form expression. Hence, our algorithm is highly efficient with the above loss functions when $\Theta$ is regularized by the squared Frobenius norm.

## 5 Empirical Results

In this section, we present our results on benchmark data sets comparing our algorithm with existing approaches in terms of generalization accuracy as well as computational efficiency. In Section 5.1, we discuss generalization results in multi-task setting. We evaluate the performance of our algorithm against several recent multi-task methods that employ clustering, low-dimensional projection of input feature space or output kernel learning. Section 5.2 discusses multi-class experiment results. Single task learning (**STL**) is a common baseline in both these experiments, and it employs hinge loss and $\epsilon$-SVR loss functions for classification and regression problems respectively. Finally, in Section 5.3, we discuss the results on the computational efficiency of our algorithm.

### 5.1 Multi-Task Data Sets

We begin with the generalization results in multi-task setups. The data sets are as follows:
**Sarcos:** A multi-task regression data set. The aim is to predict 7 degrees of freedom of a robotic arm [28].
**Parkinson:** A multi-task regression data set [29] where one needs to predict the Parkinson's disease symptom score for 42 patients.
**Yale:** A face recognition data set from the Yale face base with 28 binary classification tasks [30].
**Landmine:** A data set containing binary classification problems from 19 different landmines [30].
**MHC-I:** A bioinformatics data set having 10 binary classification tasks [12].

Table 2: Dataset statistics. $T$ represents the number of tasks and $m$ represents the average number of training examples per task.

| Dataset | $T$ | $m$ | Dataset | $T$ | $m$ |
|---|---|---|---|---|---|
| Sarcos | 7 | 15 | Landmine | 19 | 102 |
| Parkinson | 42 | 5 | MHC-I | 10 | 24 |
| Yale | 28 | 5 | Letter | 9 | 60 |
| USPS | 10 | 100 | MNIST | 10 | 100 |
| MIT Indoor67 | 67 | 80 | SUN397 | 397 | 5, 50 |

Table 3: Mean generalization performance and the standard deviation over ten train-test splits.

| Data set | STL | MTL | CMTL | MTFL | GMTL | MTRL | FMTL$_p$ $p=2$ | FMTL$_p$ $p=4/3$ | FMTL$_p$ $p=8/7$ |
|---|---|---|---|---|---|---|---|---|---|
| Regression data sets: Explained Variance (%) | | | | | | | | | |
| Sarcos | 40.5±7.6 | 34.5±10.2 | 33.0±13.4 | 49.9±6.3 | 45.8±10.6 | 41.6±7.1 | 46.7±6.9 | **50.3±5.8** | 48.4±5.8 |
| Parkinson | 2.8±7.5 | 4.9±20.0 | 2.7±3.6 | 16.8±10.8 | **33.6±9.4** | 12.0±6.8 | 27.0±4.4 | 27.0±4.4 | 27.0±4.4 |
| Classification data sets: AUC (%) | | | | | | | | | |
| Yale | 93.4±2.3 | 96.4±1.6 | 95.2±2.1 | **97.0±1.6** | 91.9±3.2 | 96.1±2.1 | **97.0±1.2** | **97.0±1.4** | 96.8±1.4 |
| Landmine | 74.6±1.6 | 76.4±0.8 | 75.9±0.7 | 76.4±1.0 | 76.7±1.2 | 76.1±1.0 | **76.8±0.8** | 76.7±1.0 | 76.4±0.9 |
| MHC-I | 69.3±2.1 | 72.3±1.9 | **72.6±1.4** | 71.7±2.2 | 72.5±2.7 | 71.5±1.7 | 71.7±1.9 | 70.8±2.1 | 70.7±1.9 |
| Letter | 61.2±0.8 | 61.0±1.6 | 60.5±1.1 | 60.5±1.8 | 61.2±0.9 | 60.3±1.4 | 61.4±0.7 | **61.5±1.0** | 61.4±1.0 |

**Letter:** A data set containing handwritten letters from several writers and having 9 binary classification tasks [31].

Table 2 presents the data set statistics. We compare the following algorithms:
**MTL** [16]: A classical multi-task learning baseline. They define the $\Theta$ matrix as: $\Theta(t, t') = \frac{1}{\mu} + \delta_{tt'}$, where $\mu > 0$ is a hyper-parameter and $\delta_{tt'} = 1$ if $t = t'$ else $\delta_{tt'} = 0$. The hyper-parameter $\mu$ is cross-validated.
**CMTL** [12]: A clustered multi-task learning algorithm. Tasks within a cluster are assumed to be close to a mean vector. It requires the number of task clusters as a hyper-parameter.
**MTFL** [11]: Learns the input kernel and the output kernel matrix as a linear combination of base kernel matrices.
**GMTL** [10]: A clustered multi-task feature learning approach. Tasks within a cluster are assumed to share a low dimensional feature subspace [2]. Hence, it effectively learns both the input kernel as well as the output kernel.
**MTRL** [9]: A multi-task relationship learning approach. It learns a low rank output kernel matrix by enforcing a trace constraint on it.
**FMTL$_p$**: Our proposed multi-task learning formulation (29). We consider three different values for the $p$-norm: $p = 2$ ($k = 1$), $p = 4/3$ ($k = 2$) and $p = 8/7$ ($k = 4$). Hinge and $\epsilon$-SVR loss functions were used for classification and regression problems respectively.

We follow the experimental protocol[1] described in [11]. Three-fold cross validation was performed for parameter selection. Linear kernel was employed for all data sets. Also, note that GMTL [10] and MTFL [11] enjoy the advantage of both input and output kernel learning. Hence, their generalization results are not directly comparable to our method, which focuses solely on learning the output kernel matrix.

Table 3 reports the performance of the algorithms averaged over ten random train-test splits. The proposed FMTL$_p$ attains the best generalization accuracy in general. It outperforms the baseline MTL as well as MTRL and CMTL, which solely learns the output kernel matrix. Moreover, it achieves an overall better performance than GMTL and MTFL. The FMTL$_{p=4/3,8/7}$ give comparable generalization to $p = 2$ case, with the additional benefit of learning sparser and more interpretable output kernel matrix (see Figure 1).

$(p = 2)$         $(p = 4/3)$         $(p = 8/7)$

Figure 1: Plots of $|\Theta|$ matrices (rescaled to [0,1] and averaged over ten splits) computed by our solver $\text{FMTL}_p$ for the Landmine data set for different $p$-norms, with cross-validated hyper-parameter values. The darker regions indicate higher value. Tasks (landmines) numbered 1-10 correspond to highly foliated regions and those numbered 11-19 correspond to bare earth or desert regions. Hence, we expect two groups of tasks (indicated by the red squares). We can observe that the learned $\Theta$ matrix at $p = 2$ depicts much more spurious task relationships than the ones at $p = 4/3$ and $p = 8/7$. Thus, our sparsifying regularizer improves interpretability.

Table 4: Mean accuracy and the standard deviation over five train-test splits.

| Data set | STL | MTL-SDCA | GMTL | MTRL | $\text{FMTL}_p$-H $p = 2$ | $p = 8/7$ | $\text{FMTL}_p$-S $p = 2$ | $p = 8/7$ | $\text{FMTL}_{kl}$ |
|---|---|---|---|---|---|---|---|---|---|
| MNIST | 84.1±0.3 | 86.0±0.2 | 84.8±0.3 | 85.6±0.4 | 86.1±0.4 | **86.2±0.4** | 82.3±0.6 | 82.4±0.3 | 82.5±0.5 |
| USPS | 90.5±0.3 | 90.6±0.2 | 91.6±0.3 | 92.4±0.2 | 92.4±0.2 | **92.6±0.1** | 87.2±0.4 | 87.5±0.3 | 87.0±0.4 |

## 5.2 Multi-Class Data Sets

The multi-class setup is cast as $T$ one-vs-all binary classification tasks, corresponding to $T$ classes. In this section we experimented with two loss functions: a) $\text{FMTL}_p$-H – the hinge loss employed in SVMs, and b) $\text{FMTL}_p$-S – the squared loss employed in OKL [17]. In these experiments, we also compare our results with **MTL-SDCA**, a state-of-the-art multi-task feature learning method [32]. In addition, we report results from our KL-divergence regularized formulation with squared loss (denoted by $\text{FMTL}_{kl}$ ).

**Handwritten Digit Recognition**: We consider the following two data sets and follow the experimental protocol detailed in [10].
**USPS**: A handwritten digit data sets with 10 classes [33]. We process the images using PCA and reduce the dimensionality to 87. This retains almost 87% of variance.
**MNIST**: Another handwritten digit data set with 10 classes [34]. PCA is employed to reduce the dimensionality to 64.

We use 1000, 500 and 500 examples for training, validation and test respectively. Table 4 reports the average accuracy achieved by various methods on both data sets over 5 splits. Our approach $\text{FMTL}_p$-H obtains better accuracy than GMTL, MTRL and MTL-SDCA [32] on both data sets.

**MIT Indoor67 Experiments**: We also report results on the MIT Indoor67 benchmark [35] which covers 67 indoor scene categories with over 100 images per class. We use the train/test split (80/20 images per class) provided by the authors. $\text{FMTL}_p$-S achieved the accuracy of 73.1%, 73.1% and 73.3% with $p = 2, 4/3$ and $8/7$ respectively. Our KL-divergence regularized approach $\text{FMTL}_{kl}$ obtained 73.1%. Note that these are better than the ones reported in [36] (70.1%) and [35] (68.24%).

**SUN397 Experiments**: SUN397 [37] is a challenging scene classification benchmark [35] with 397 scene classes and more than 100 images per class. We use $m = 5, 50$ images per class for training, 50 images per class for testing and report the average accuracy over the 10 standard splits. We employed the CNN features extracted with the convolutional neural network (CNN) provided by [35] using Places 205 database. We resized the images directly to $227 \times 227$ pixels, which is the

Table 5: Mean accuracy and the standard deviation over ten train-test splits on SUN397.

| $m$ | STL | MTL | MTL-SDCA | FMTL$_p$-H | | | FMTL$_p$-S | | | FMTL$_{kl}$ |
|---|---|---|---|---|---|---|---|---|---|---|
| | | | | $p=2$ | $p=4/3$ | $p=8/7$ | $p=2$ | $p=4/3$ | $p=8/7$ | |
| 5 | 40.5±0.9 | 42.0±1.4 | 41.2±1.3 | 41.5±1.1 | 41.6±1.3 | 41.6±1.2 | **44.1±1.3** | **44.1±1.1** | 44.0±1.2 | **44.1±1.3** |
| 50 | 55.0±0.4 | 57.0±0.2 | 54.8±0.3 | 55.1±0.2 | 55.6±0.3 | 55.1±0.3 | **58.6±0.1** | 58.5±0.1 | **58.6±0.2** | 58.4±0.1 |

$(p=2)$      $(p=4/3)$      $(p=8/7)$

Figure 2: Plots of matrices $\log(1+|\Theta|)$ (rescaled to [0,1] and diagonal entries removed since they reflect high similarity of a task with itself, which is obvious) computed by our solver FMTL$_p$-S for the SUN397 data set for different $p$-norms, with cross-validated hyper-parameter values. The hierarchical block structure indicated by the red squares corresponds to the groups of classes available in SUN397, e.g., the top 3 super-classes are *indoor*, *outdoor-natural*, and *outdoor-man-made*, which in turn contain subgroups of classes. Note that this information was not used in experiments. We can observe that the learned $\Theta$ matrix at $p=2$ depicts much more spurious task relationships than the one at $p=8/7$. Thus, our sparsifying regularizer improves interpretability. Best viewed in color.

size of the receptive field of that network. The parameters were set by 2-fold cross-validation. The results are tabulated in Table 5.

Figure 2 offers a qualitative assessment of the proposed method by showing the output kernel matrices $\Theta$ computed by our formulation FMTL$_p$-S for various $p$-norms. We can observe that the $\Theta$ matrix becomes sparser as the $p$-norm decreases from 2 towards one. Enforcing sparsity helps to detect the hierarchical structure of the tasks (see caption for more details).

(a)      (b)

Figure 3: (a) Plot compares the runtime of various algorithms with varying number of tasks on SUN397. Our approach FMTL$_2$-S is 7 times faster that OKL [17] and 4.3 times faster than ConvexOKL [18] when the number of tasks is maximum. It can be observed that FMTL$_2$-S also has the best computational complexity in terms of number of tasks. (b) Plot showing the factor by which FMTL$_2$-S outperforms OKL and ConvexOKL over the hyper-parameter range on various data sets. On SUN397, we outperform OKL and ConvexOKL by factors of 5.2 and 7 respectively. On MIT Indoor67, we are better than OKL and ConvexOKL by factors of 8.4 and 2.4 respectively.

Figure 4: (a)Plot showing the factor by which $\text{FMTL}_2\text{-S}$ outperforms OKL and ConvexOKL over the hyper-parameter range on MNIST and USPS data sets. On MNIST, we outperform OKL and ConvexOKL by factors of $25.5$ and $6.3$ respectively. On USPS, we are better than OKL and ConvexOKL by factors of $26.2$ and $7.4$ respectively. (b) Plot comparing the average rate at which the three algorithms achieve the optimal primal objective (on SUN397 data set). $\text{FMTL}_2\text{-S}$ was run with a duality gap of $10^{-15}$ and its primal objective value was taken to be the optimal primal objective.

## 5.3 Scaling Experiment

We compare the runtime of our solver for $\text{FMTL}_2\text{-S}$ with the OKL solver of [17] and the Convex-OKL solver of [18] on several data sets. All the three methods solve the same optimization problem. Figure 3a shows the result of the scaling experiment where we vary the number of tasks (classes). The parameters employed are the ones obtained via cross-validation. Note that both OKL and ConvexOKL algorithms do not have a well defined stopping criterion whereas our approach can easily compute the relative duality gap (set as $10^{-3}$). We terminate them when they reach the primal objective value achieved by $\text{FMTL}_2\text{-S}$ . Our optimization approach is 7 times and $4.3$ times faster than the alternate minimization based OKL and ConvexOKL, respectively, when the number of tasks is maximal. The generic $\text{FMTL}_{p=4/3,8/7}$ are also considerably faster than OKL and ConvexOKL.

Figure 3b compares the average runtime of our $\text{FMTL}_p\text{-S}$ with OKL and ConvexOKL on the cross-validated range of hyper-parameter values. The hyper-parameter value chosen by cross-validation for SUN397 and MIT Indoor67 data sets was around $10^5$. $\text{FMTL}_p\text{-S}$ outperform them on both MIT Indoor67 and SUN397 data sets. Figure 4 shows the same comparison on MNIST and USPS data sets.

## 6 Conclusion

We proposed a novel formulation for learning the positive semi-definite output kernel matrix for multiple tasks. Our main technical contribution is our analysis of a certain class of regularizers on the output kernel matrix where one may drop the positive semi-definite constraint from the optimization problem, but still solve the problem optimally. This leads to a dual formulation that can be efficiently solved using stochastic dual coordinate ascent algorithm. Results on benchmark multi-task and multi-class data sets demonstrates the effectiveness of the proposed multi-task algorithm in terms of runtime as well as generalization accuracy.

**Acknowledgments.** P.J. and M.H. acknowledge the support by the Cluster of Excellence (MMCI).

## Footnotes

[1]The performance of STL, MTL, CMTL and MTFL are reported from [11].