[Reviews · NeurIPS 2015]

Submitted by Assigned_Reviewer_1

The present paper focus on kernel-based multi-task learning. More specifically, it deals with learning the output kernel which defined over multiple tasks. Starting from the vector-valued RKHS formulation of the multi-task learning problem, the authors propose an output kernel learning algorithm that are able to learn both the multi-task learning function and task dependencies. The authors first provide an optimization formulation of the problem which is jointly convex. Then they show that this optimization can be solved without the positive-semidefiniteness constraint

of the output kernel, which is computationally costly, and propose the use of a stochastic dual coordinate ascent method to solve it.

Experiments on multi-task data sets and comparison in terms of performance and running time with previous multi-task learning algorithms are provided.

The paper builds upon recent studies on multitask learning and output kernel learning. In particular, it is closely related to: [1] Dinuzzo & al., Learning output kernels with block coordinate descent., ICML 2011. [2] Zhang & Yeung, A convex formulation for learning task relationships in multi-task learning, UAI 2010.

While these two references are clearly mentioned in the paper, there are other closely related papers which are not discussed and cited. For example [3] Kadri & al., Multiple operator-valued kernel learning, NIPS 2012. [4] Scalable matrix-valued kernel learning for high-dimensional nonlinear multivariate regression and Granger causality, UAI 2013. [5] Ciliberto & al., Convex learning of multiple tasks and their structure, ICML 2015.

These papers, in particular [4,5], needs to be cited and discussed. The proposed approach needs to be compared to them from a theoretical and experimental perspectives.

The contribution is somewhat limited. The main novelty is the dual formulation of the optimization problem which does not need the positive-semidefiniteness constraint of the output kernel. However, it should be noted that even in the previous works [1] and [2] where an alternating optimization scheme is adopted, positive-semidefiniteness constraint is not needed. Incorporating the solution of the minimization over the multi-task learning function in the optimization problem of the output kernel learning (without the positive-semidefiniteness constraint) provides a positive semi-definite matrix over tasks. This is clearly mentioned in [1] (see Eq. 7) and [2]. The way this paper is written gives the impression that it is the first that avoid the optimization over S_+^\top.

Also, the result about the joint convexity of the output kernel learning optimization problem is not new. This is largely discussed in [5] and also in [2]. It is true that [5] is a newly published paper but the contribution of the present work

will be more visible by taking into account recent results in this field.

In the experiments, it is not clear why the proposed approach is compared to OKL in terms of running time and not in terms of performance. The paper is about output kernel learning, so the performance of the proposed method needs to be compared to those of [1, 4, 5].

update: The authors did a good job in answering my questions. My main concern is about the presentation of the results and their novelty. I think that is technical novelty and originality in the proposed approach, however the authors should better explain this and make it clear what the novel contributions of the paper are in comparison to what is known.
Summary: The paper provides an output kernel learning algorithms that avoid the positive-semidefiniteness constraint of the output kernel. The paper provides some interesting results and a new formulation of the output kernel learning problem. The writing can be improved. The contribution is somewhat limited. Some closely related references are missing.

Submitted by Assigned_Reviewer_2

This is a well-written paper that introduces a novel method for multi-task learning. The mathematical derivation of the resulting optimization problem and the algorithm used to estimate the parameters are well explained. The usefulness is supported by several experiments on multi-task learning and multi-class classification problems.

However, I have one major concern w.r.t. the method that is introduced. Despite the good performance in experiments, I am not really convinced of the formulation in Eqns. (2) and (3). Those formulations do not allow for a separate regularization mechanism for tasks and parameter vectors of a single task, respectively. More specifically, it is known in the field that two different regularization mechanisms have different effects in multi-task learning. Adding a regularizer for the parameters of a single task prevents overfitting, as in traditional classification or regression settings.

However, including a second regularization mechanism that shrinks models, so that related tasks have similar models, is known to improve the predictive performance (see several references from the reference list). Many existing methods for multi-task learning adopt this principle, in which two different regularization parameters are tuned in a validation step. The method of the authors does not have this flexibility, so theoretically I would expect that it is inferior w.r.t. predictive performance, compared to existing methods. The issue is not at all discussed in the paper, and I think this is an important point. It should be discussed why formulation (2) is a particularly good approach for tackling multi-task learning problems.

Summary: This is an interesting paper, but I have one major concern about the proposed methodology (see below).

Submitted by Assigned_Reviewer_3

The main contribution lies in Section 3, where an efficient optimization procedure is presented to solve a convex subproblem on a PSD cone. Some notations are confusing. For example, the variable 'k' is used to denote the kernel function in Section 2, but in Section 3 it is also used to define p in Theorem 3 and define a subscript in Theorem 4. So it is better to use different symbols in different places.

In the experiments on the SUN397 dataset, the authors are encouraged to compare with other MTL methods instead of just the MTL-SDCA method.
Summary: This paper presents a new optimization method for some existing works [7,14] when using some appropriate regularizers for a task covariance matrix.

Author Feedback
Author rebuttal: We thank all reviewers for their feedback. We highlight our main contributions

- using the theory of positive definite kernels we show that the constraint that the output kernel is positive semi-definite (psd) can be dropped in our multi-task learning formulation and derive an unconstrained dual optimization problem (line 212) which we solve efficiently via dual coordinate ascent. We have an average speed-up of 7-25 compared to OKL depending on the dataset. We characterize the class of regularizers for which dropping the psd constraint is possible (Theorem 5). This is a highly original contribution and could also lead to more efficient algorithms in other domains.

- we show that the p-norm regularizer for the output kernel with small p leads to better prediction performance (Table 3+4) and also better interpretability of the output kernel (Figure 1) as spurious correlations between tasks are eliminated.

Detailed Comments:

References [R1-5] below are as used by Reviewer 1. Rest of the references are as in the paper.

Reviewer 1

- "contribution is limited as .... previous work [R1] and [R2] need not the positive semi-definiteness constraint"

[R1,R2,R5] have indeed closed form expressions for the output kernels in their alternating schemes for fixed weights. However, computation of these expressions requires a costly eigenvector decomposition (EVD) (cost O(T^3)). [R1] needs to solve a Sylvester equation which is solved most efficiently using an EVD (the authors of [R1] implement it like this in their code).
In contrast, we show that the dual optimization problem is unconstrained and the output kernel can be obtained by applying *elementwise* a function to all entries of a certain matrix which requires *no* EVD and is automatically positive semi-definite. Via the primal-dual gap we have then a well-defined stopping criterion in contrast to the alternating schemes.

- "the joint convexity of the output kernel learning problem is not new"

We are nowhere claiming novelty. The main step for the joint convexity is Lemma 1 for which we cite [2, R2]. In 114-119 we discuss similarities resp. differences to existing work. However, the complete derivation in this explicit form is not contained anywhere. While [R5] is similar (this paper appeared just a few days before the NIPS deadline), we think that our derivation avoiding vector-valued RKHS is easier to understand.

- "why is the proposed approach compared to OKL [R1] in terms of running time but not in performance, performance needs to compared to [R4,R5]"

For squared loss and Frobenius norm regularizer on Theta, OKL is the same as our approach denoted FMTL_2-S (p=2 and S for squared loss). As the problem is convex, our and their solver yield the same result.
We got the code from the authors of [R5]. While for the OKL/FMTL_2-S formulation, our FMTL_2-S is 4.3 times faster than [R5] with the best hyper-parameter setting and on avg 7 times faster on all hyper-parameter values (on SUN dataset). An updated Fig.2 with timings of [R5] is available at:
http://postimg.org/image/svdw70mcl/
Regarding prediction we compare with all the formulations studied in [R5], which are: CMTL [9],OKL [R1],MTRL [R2] and [2]. [2] is a special case of GMTL[8] with 1 group and we have included this case in our experiments.
The approach of [R4] learns an input and output kernel. We compare prediction performance with other formulations that learn input and output kernels (MTFL[23], GMTL[8], see line 314/315). Moreover, the authors of [R4] told us that code for their method is not available.

Reviewer 3

- The proposed formulation should have a separate regularization mechanism for every task. It is known to improve predictive performance

In principle, we agree. We implicitly have a separate regularization for every task via the output kernel (Eq (4) and line 144). If the output kernel is a diagonal matrix, then the inverse diagonal entries correspond to task-specific regularization parameters. Thus we tune implicitly task specific regularization parameters when learning the output kernel. Implicit regularization has been used in previous multi-task approaches [14,23]. We also compare to approaches which explicitly regularize every task [R2,9,13] and outperform them.

Reviewer 4

- for SUN397 dataset, the authors should compare with other MTL methods instead of just MTL-SDCA

MTL-SDCA was chosen as they have state-of-the-art results on SUN397 dataset and have a scalable multiclass implementation. In the meantime we have efficiently implemented MTL [13]. Accuracies are: 42.0(m=5),47.4(m=10),51.7(m=20),57.0(m=50) for SUN
(we are better). The implementation of MTRL [R2] is not efficient for the multi-class setting which leads to memory problems for SUN. We are implementing a memory-optimized version of MTRL (for multiclass) and will report its results in the final version. Please note that for USPS/MNIST we compare to GMTL and MTRL.